# Apparent symmetry rising induced by crystallization inhibition in ternary co-crystallization-driven self-assembly

Siyu Xie [1,2], Wenjia Sun[1], Junliang Sun [1], Xinhua Wan [1,2] & Jie Zhang [1,2] ✉

The concept of apparent symmetry rising, opposite to symmetry breaking, was proposed to illustrate the unusual phenomenon that the symmetry of the apparent morphology of the multiply twinned particle is higher than that of its crystal structure. We developed a unique strategy of co-crystallization-driven self-assembly of amphiphilic block copolymers PEO-*b*-PS and the inorganic cluster silicotungstic acid to achieve apparent symmetry rising of nanoparticles under mild conditions. The triangular nanoplates triply twinned by orthogonal crystals (low symmetry) have an additional triple symmetry (high symmetry). The appropriate crystallization inhibition of short solvophilic segments of the block copolymers favors the oriented attachment of homogeneous domains of hybrid nanoribbons, and consequently forms kinetic-controlled triangular nanoplates with twin grain boundaries.

As a basic attribute widely existing in geometry, informatics, crystallography, physics, etc., symmetry is common but not always maintained. High symmetry refers to more numbers of symmetry elements contained in the system, strictly speaking, the order of the symmetric group. Symmetry breaking that symmetry evolves from high to low, is usual in nature, such as energy non-conservation in general relativity, parity non-conservation, phase transition, and chirality. The reversed evolution way, symmetry rising, conforms to the interesting concept of emergence phenomenon in systematics, i.e., a complex system emerges with additional high symmetry that does not belong to any constituent. Up to date, only a few examples of symmetry rising in crystallography were found, such as 5-fold twinned particles of face-centered cubic materials caused by defects[1-3], even though structural defects caused by kinetic disturbance had been usually considered to lead to symmetry breaking of crystals. We propose apparent symmetry rising (ASR) to describe that the symmetry of the apparent morphology of a multiply twinned particle is higher than that of its microscopic unit cell. ASR endows nanoparticles with unique properties, such as improved mechanical performance[4,5], low thermal conductivity[6], and striking optical properties[7,8]. However, the realization of ASR is limited to the metal or alloy materials with a face-centered cubic cell prepared under severe processing condition, ball milling[9] or high-pressure torsion[10,11], which is critical to induce plastic deformation. It is still a

challenge to fabricate feasible strategies to achieve ASR under mild conditions for better understanding and application of symmetry rising.

Crystallization-driven self-assembly (CDSA) of amphiphilic block copolymers with a crystalline core-forming block is emerging as a promising method to construct well-defined nanostructured particles[12,13]. The fabrication methods of crystals, such as seeded growth[14,15] and self-seeding[16,17], have been successfully applied to CDSA to tune the nanostructures. Not surprisingly, the symmetry of the CDSA assemblies reported in the literature usually coincides with that of the unit cells of crystalline cores[18,19]. The observed defects in crystalline assemblies formed from some organic polymers with weak crystallinity break the apparent symmetry of the assemblies[20-22]. In an alternative way, crystalline inorganic components which form eutectics with organic polymers, can be introduced into CDSA to improve the crystallinity of the hybrids and thus reduce the probability of symmetry breaking[23-25]. On the other hand, notably, the solvophilic blocks covalently linked with crystalline ones in the block copolymers play a critical role in stabilizing the surface of crystals and tuning the morphology in CDSA, which is analogous to surfactants in the synthesis of inorganic nanocrystals[26]. By changing the chemical structure of solvophilic segments, CDSA can flexibly realize the morphological evolution of 1D[27,28], 2D[29,30], and chiral[31,32] core-crystalline assemblies.

[1]Beijing National Laboratory for Molecular Sciences, College of Chemistry and Molecular Engineering, Peking University, 100871 Beijing, China. [2]Key Laboratory of Polymer Chemistry and Physics of Ministry of Education, Peking University, 100871 Beijing, China. ✉e-mail: jz10@pku.edu.cn

We posit that the solvophilic segments would also behave as crystallization inhibitors, similar to the role of surfactants[33], which could introduce grain boundary defects and induce twinning, and eventually may be favorable to induce ASR.

Herein, we illustrate a strategy that can realize ASR mildly by crystallization inhibition induction via co-crystallization-driven self-assembly (CCDSA) of amphiphilic block copolymers and the inorganic clusters in solution. The inorganic cluster silicotungstic acid $H_4SiW_{12}O_{40}$ (STA) as a strong acid can protonate the oxygen atoms in poly(ethylene oxide) (PEO) and the STA/PEO form eutectics in situ in selective solution in the form of intimate ion pairs through electrostatic interaction[34]. The block copolymer polystyrene-*b*-poly(ethylene oxide) (PEO-*b*-PS) can co-crystallize with STA to self-assemble into the STA/PEO eutectic core (Fig. 1a). Besides stabilizing the crystalline core, the short solvophilic PS as crystallization inhibitors could dope into the eutectic as defects during crystallization, aid to form twin grain boundaries, and consequently induce ASR.

## Results

### Co-crystallization-driven self-assembly

A series of $PEO_{45}$-*b*-$PS_n$ samples with various molecular weights of PS were synthesized, and their structures were confirmed by $^1H$ NMR spectroscopy and gel permeation chromatography (GPC) (Supplementary Fig. 1, 2, Supplementary Table 1). For an example of a typical CCDSA experiment, STA in THF solution containing 1% vol water was gradually added to the 1:1 mixed solution of $PEO_{45}$ and $PEO_{45}$-*b*-$PS_{16}$ in THF/DCM, and the solution became turbid immediately. After standing for 24 h, triangular nanoplates with sizes of ~500 nm were observed under bright-field transmission electron microscopy (TEM) (Fig. 1b) and scanning electron microscopy (SEM) (Fig. 1c). In comparison, no assembly occurred for any individual component in the same mixed solvents, demonstrating that the triangular nanoplates were in situ formed from ternary complexes in solution.

The corresponding energy-dispersive X-ray spectrum (EDS) mapping of the W and Si elements in STA and the S element in the 2-dodecylsulfanylthiocarbonylsulfanyl-2-methyl-propanate (DMP) -ended PS chains (Fig. 1g–i, Supplementary Fig. 3) match well with the positions of the nanoplates in the high-angle annular dark field scanning transmission electron microscopy (HAADF-STEM) image (Fig. 1f). The concentrated S contents in the region of triangular nanoplates imply that DMP-ended PS chains cover the surface so that the block polymer chains are aligned perpendicularly to the surface of the triangular nanoplates. The average height of triangular nanoplates was statistically analyzed to be ~50 nm based on atomic force microscope (AFM) (Fig. 1d, e, Supplementary Fig. 4). Considering that the shell of collapsed PS is relatively thin, the proximate thickness of crystalline cores of STA/PEO is ~50 nm. As the length of the extended $PEO_{45}$ is estimated to be ~16 nm, the crystalline cores of the triangular

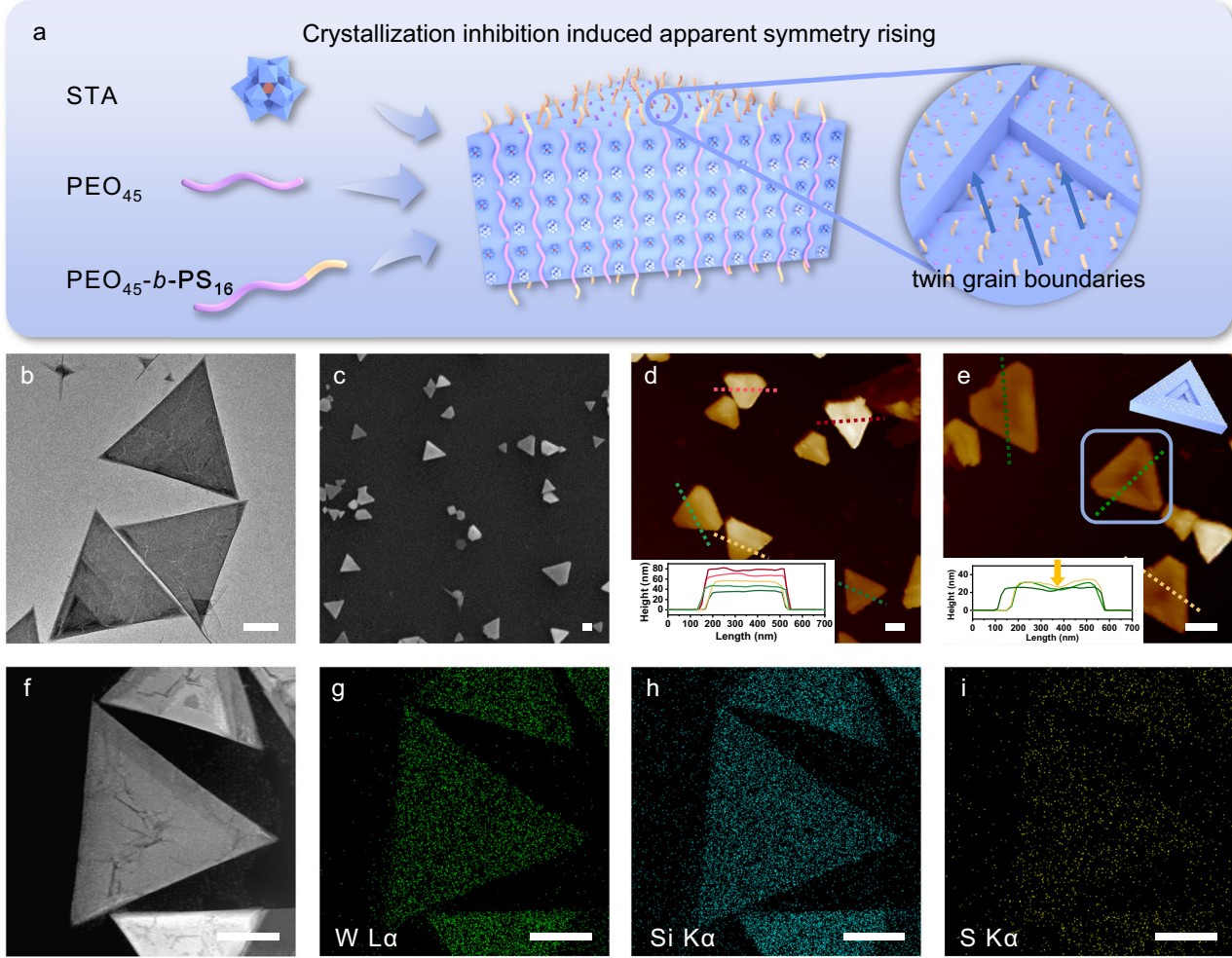

**Fig. 1 | Schematic representation and morphological characterization of the triangular nanoplates obtained by co-crystallization-driven self-assembly (CCDSA). a** Schematic representation of CCDSA of STA/PEO/PEO-*b*-PS. TEM (**b**), SEM (**c**) and AFM (**d**, **e**) images of STA/PEO/PEO-*b*-PS triangular nanoplates prepared by CCDSA. Inset diagrams show height profiles along the lines in the AFM images. **f** A HAADF-STEM image of triangular nanoplates. Corresponding EDS mapping of W (**g**), Si (**h**), and S (**i**). All scale bars are fixed to 200 nm.

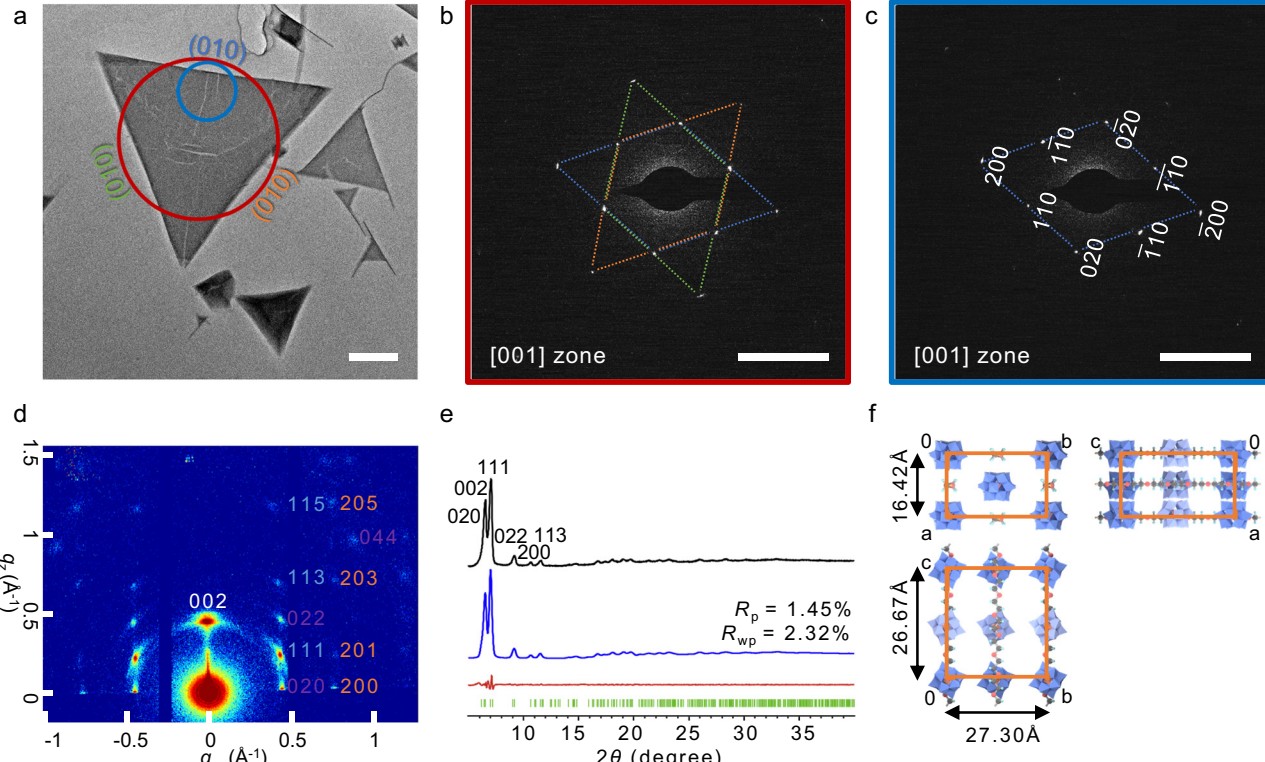

**Fig. 2 | Analysis and refinement of crystal structure of triangular nanoplates.**
**a** A TEM image of the triangular nanoplates and corresponding SAED patterns taken from overall (**b**) and local (**c**) selected regions along the [001] direction. **d** 2D GIWAXS pattern of the triangular nanoplates. **e** PXRD profiles of the experimental pattern (black curve), Le Bail fitting pattern (blue curve), their difference (red curve), and Bragg position (green curve). **f** Schematic representation of STA/PEO crystal structure. The blue octahedron represents {WO₆}, the orange tetrahedron represents {SiO₄}, the red spheres represent oxygen atoms, the black spheres represent carbon atoms, and the cyan spheres represent hydrogen atoms. The scale bars are 200 nm for (**a**) and 1 nm⁻¹ for (**b**, **c**).

nanoplates may contain ~3 layers of PEO₄₅ chains, i.e. a sandwich structure with the PEO/PEO-*b*-PS in the external layers and the PEO homopolymer in the inner core.

## Crystal structure

The crystal structure of the triangular nanoplates was first examined by selected area electron diffraction (SAED). The overall SAED exhibits a hexagonal pattern consistent with the triple apparent symmetry of the triangle (Fig. 2a, b). Whereas, the local SAED around the edge of the triangle shows an orthogonal pattern with fewer diffraction spots (Fig. 2c). The overall SAED pattern along the [001] zone can be decomposed into three sets of the local orthogonal patterns connected at 60°, indicating that the triangular nanoplates are pseudo-merohedral twins with twin-lattice quasi-symmetry[35], so the local SAED pattern was used to analyze the unit cell parameter. The odd *h*00 and 0*k*0 diffractions are extinct along the [001] zone. The SAED pattern along the [011] zone with a tilted angle of 44° around the a*-axis was obtained to determine the c-axis dimension, at which the 022 and the 111 diffraction spots appear (Supplementary Fig. 5).

The cell parameters were further confirmed by grazing-incidence wide-angle X-ray scattering (GIWAXS) (Fig. 2d). As the triangular nanoplates were deposited in a face-on orientation on the substrate under AFM observation, their c-axis were perpendicular to the substrate with different XY-plane orientations. The diffractions of (100), (010), and (110) crystal planes, coincide at the 002 diffraction spot. The *hkl* diffraction spots with the same *l* value are arranged along the same horizontal line. The reflection condition can be determined as *h*00: *h* = 2*n*; 0*k*0: *k* = 2*n*; *hk*0: *h* + *k* = 2*n* from SAED, and 00*l*: *l* = 2*n*; 0*kl*: *k* = 2*n*; *h*0*l*: *l* = 2*n* from GIWAXS, indicating the space groups of *C*222₁ (no. 20). The powder X-ray diffraction (PXRD) (Fig. 2e) in the range of

2*θ* from 5° to 40° displays strong signals at low angles labeled with diffraction indexes and weak broad ones at high angles. The Le Bail fitting shows the unit cell of a = 1.6420(6) nm, b = 2.7297(7) nm, c = 2.6668(8) nm, α = 90°, β = 90°, γ = 90° in the space group of *C*222₁ with a final $R_{wp}$ of 2.32% and $R_p$ of 1.45% (Supplementary Table 2), which further confirms the experiment results. The simulated XRD profile and SAED pattern along the [001] direction for the STA/PEO crystal structure (Supplementary Fig. 6) is in good agreement with the experimental ones.

As the orthorhombic cell parameters of triangular nanoplates have a relationship of $b \approx \sqrt{3}a$, the lattices after 60° rotation can overlap well with the original one. The sphere-approximate shape and delocalized charges of STA make little difference between STAs of different orientations. The above two factors are beneficial to twinning by pseudo-merohedry. The orthorhombic unit cell parameters are significantly larger than those of STA crystals by SAED and PXRD analysis (Supplementary Fig. 7), indicating that polymers penetrate through the lattice of STA crystals. According to the literature, PEO may be distributed in the interstitial channels between STAs with a straight or helical conformation[36–38], and the PEO chain should be aligned in parallel to the c-axis of the unit cell according to the EDS results. Consequently, it can be concluded that the crystalline cores of the triangular nanoplates are 3-fold pseudo-merohedral twins generated from orthorhombic cells and the schematic representation of the crystal structure was displayed in Fig. 2f.

The triangular nanostructures are not frequently observed in CDSA, and the crystal phase structure is completely different from the previous reported triangles. The triangular nanostructures, whether formed by face-centered cubic metals[39–44] or PLA stereo-composite crystals[45–48], are single crystal nanostructures, but the triangular

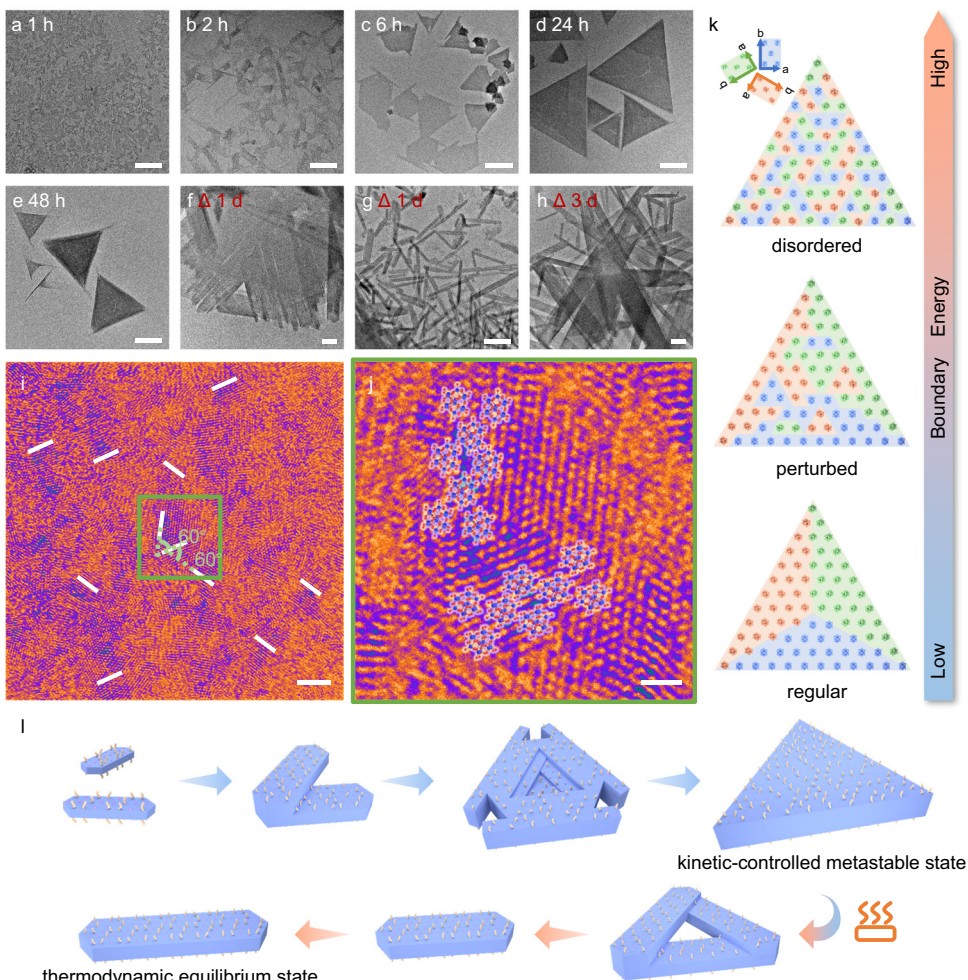

**Fig. 3 | Mechanism for ASR of triply twinned particles. a–e** TEM images of the growth process of triangular nanoplates within 48 hours. **f–h** TEM images of collapsed triangular nanoplates after annealing for 1~3 days. Pseudo-color HRTEM images of the triangle precursor with orientation labels (**i**) and its partial magnified images with schematical representations of STA (**j**). **k** Schematic illustrations of the internal STA arrangement in the triangular nanoplate. **l** Schematic representation of the growth of triangular nanoplates and their collapse after annealing. The scale bars are 200 nm for (**a–h**), 3 nm for (**i**), and 1 nm for (**j**).

nanoplates in the present work are essentially a special kind of twins—pseudo-merohedral twins.

**The mechanism of apparent symmetry rising**

Such an unprecedented feature that the triple symmetry of mesoscopic crystalline self-assemblies is higher than the orthogonal symmetry of microscopic unit cell structure is a type of ASR. To investigate the mechanism of the ASR formation, the kinetic features at various times during the CCDSA process were tracked (Fig. 3a–e). After mixing solutions for 1 h, small nanoribbons of tens of nanometers formed, some of which were connected at 60°. At 2 h, V-shaped or triangular precursors of ~100 nm assembled by orientated nanoribbons can be observed. The triangular precursors of ~200 nm with obvious unclosed angles and epitaxial growing edges occurred within 6 h. The precursors gradually grew and became complete triangles over time. Finally, a large amount of regular triangular nanoplates were obtained at 24 h, and the size and shape remained unchanged within 48 h.

Afterward, the as-prepared triangular nanoplates were annealed at 80 °C in the sealed solution. After standing for 1 day, the original regular triangular nanoplates almost disappeared, and a large number of nanoribbons can be observed (Fig. 3f, g). After 3 days, the nanoribbons elongated significantly, and a few of nanoribbons continued to grow on the partially dissolved triangular framework (Fig. 3h). The higher the annealing temperature, the faster the disassembly of

triangular nanoplates (Supplementary Fig. 8). Annealing at high temperature makes the triangular nanoplates disassembled and slowly recrystallized to form thermodynamically stable nanoribbons with fewer defects. Further EDS mapping and SAED indicate that the unit cell structure of the nanoribbons is consistent with that of the triangular nanoplates (Supplementary Fig. 9). The nanoribbons are thermodynamic-controlled single crystals, the morphologies of which are similar to those of the annealed hybrid crystals in the thin film of STA and PEO or its block copolymers in literature[23,24]. The above results suggest that the triangular nanoplates are constructed from the oriented attachment of the nanoribbons and are kinetic-controlled assemblies.

The oriented attachment structures play an important role in ASR of the triangular nanoplates. The local oriented substructures of the triangle precursor at 6 h were investigated by high-resolution TEM (HRTEM) (Fig. 3i, j, Supplementary Fig. 10). The purple regions with clear lattice patterns are the exposed crystal face formed by $PEO_{45}$ and STA, and the orange regions with blurred lattice patterns are the PS-shielded domains. The lattice points representing W atoms in STA can match well with the analysis result of crystal structure. Due to the rotation of STA along the a-axis as shown in the schematic representation of STA/PEO (Fig. 2f), the projection of W atoms on the (001) crystal plane formed a beaded line parallel to the [010] lattice direction (marked as a white line), which can be used to resolve the local

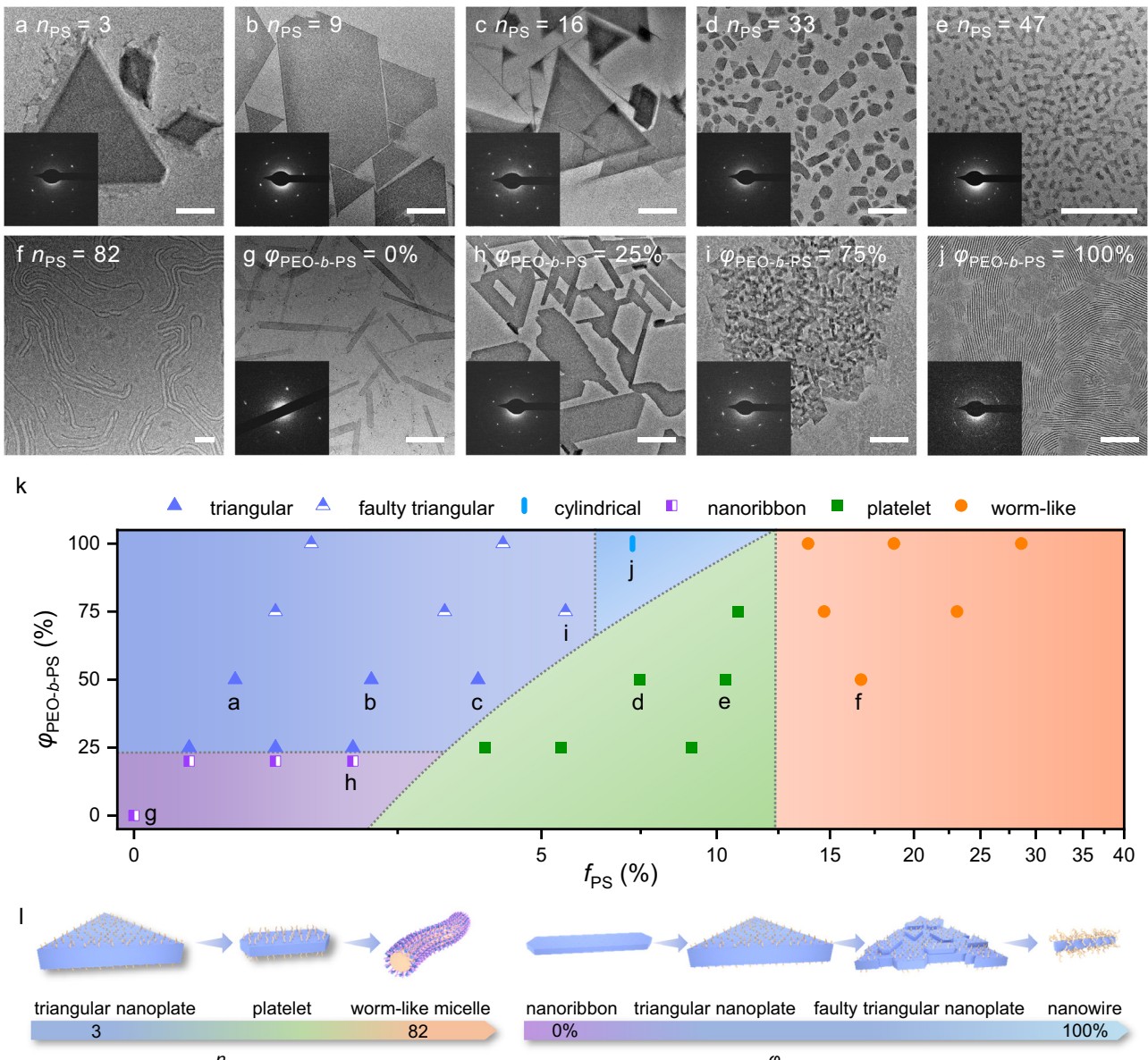

**Fig. 4 | Morphological evolution and phase diagram at various volume fractions of PS ($f_{PS}$) and molar ratio of PEO-*b*-PS to total polymers ($\varphi_{PEO-b-PS}$).** **a–f** TEM images of assemblies of STA/PEO$_{45}$/PEO$_{45}$-*b*-PS$_n$ at the fixed $\varphi_{PEO-b-PS}$ ~ 50%. **g–j** TEM images of assemblies of STA/PEO$_{45}$/PEO$_{45}$-*b*-PS$_{16}$ at various $\varphi_{PEO-b-PS}$. The insets: the corresponding SAED. **k** The phase diagram of the ternary assemblies at various $f_{PS}$ and $\varphi_{PEO-b-PS}$, which is divided into monocrystalline (purple and green region), polycrystalline (blue region), and amorphous phases (orange region). **l** Schematic representations of morphological evolution with increasing PS length and $\varphi_{PEO-b-PS}$. All scale bars are fixed to 200 nm.

oriented structure of the crystalline core. The orientation of neighboring domains near the PS-shielded domains maintains a constant joint angle of 60°, which indicates that the crystallization inhibition of PS leads to stacking faults. In comparison, the STA/PEO assemblies without PS are nanoribbons with homogeneous orientations (Fig. 4g). As another evidence of the oriented attachment, depressions can also be observed at the center of triangular nanoplates (Fig. 1e, Supplementary Figs. 4, 11). Considering that PEO-*b*-PS is involved in the early stage of crystallization to induce the oriented attachment, the PS segment may inhibit further vertical growth, especially at the interior of the triangle framework.

In general, the formation of twinned particles should be energetically favored, i.e., maximization of homogeneous domains and minimization of grain boundaries. According to HRTEM images, the triangular nanoplates would have perturbed crystalline cores, the boundary energy of which is between those of regular and disordered

crystalline cores (Fig. 3k). The homogeneous domains cannot be maintained at large areas due to PEO$_{45}$ and PEO$_{45}$-*b*-PS$_{16}$ randomly distributing on the crystal surface. A different orientation domain deviating from the original orientation would form near the local domain with relatively enriched PEO$_{45}$-*b*-PS$_{16}$. It can be conclusive that the crystallization inhibition of amorphous PS is critical to ASR.

**Morphological evolution tuned by crystallization inhibition**
Since the crystallization inhibition of PS can promote the formation of ASR, the effect of the crystallization inhibition strength on the morphological evolution of the assemblies can also be further studied by varying volume fractions of PS ($f_{PS}$) and the molar ratio of PEO-*b*-PS to total polymers ($\varphi_{PEO-b-PS}$) (Supplementary Methods). First, $\varphi_{PEO-b-PS}$ was fixed at 50%, i.e., the molar ratio 1:1 of PEO$_{45}$ to PEO$_{45}$-*b*-PS$_n$. The triangular nanoplates kept unchanged until the polymerization degree $n$ of PS increased to $n = 16$ ($f_{PS} = 3.76\%$) (Fig. 4a–c). As the length of PS

increases to $n = 33$ ($f_{PS} = 7.47\%$), the expanded PS-shielded domains are unfavorable to form orientation attachments, and thus monocrystalline platelets with comparable sizes of {010} and {110}, much smaller than triangular nanoplates, were generated (Fig. 4d). As for the PS block with $n = 47$ ($f_{PS} = 10.31\%$), the growth of platelets was severely inhibited, and the sizes became small (Fig. 4e). The crystallization process is completely inhibited in the case of PEO$_{45}$-$b$-PS$_{82}$ ($f_{PS} = 16.7\%$). The microphase separation due to the amphiphilicity of block copolymers enables the formation of reversed worm-like micelles which are similar to previous reports (Fig. 4f)[49,50]. The ternary assemblies underwent a phase transition from triangular nanoplates to platelets and then to reversed worm-like micelles with an increase of $f_{PS}$ (Fig. 4l). In addition, the molar ratio of STA to total PEO and their respective concentration were also varied to tune the morphology at the fixed $\varphi_{PEO\text{-}b\text{-}PS}$ of 50%. In general, triangular nanoplates can be formed in a wide molar ratio range, and the size and shape regularity of the morphology decrease with the decrease of the concentration (Supplementary Fig. 12). Moreover, by varying the length of PEO segment, the hybrid assemblies based on the block copolymer PEO$_{113}$-b-PS$_{30}$ ($f_{PS} = 5.52\%$) show similar morphology and SAED patterns to those of PEO$_{45}$-b-PS$_{16}$ (Supplementary Fig. 13).

The morphological evolution of the assemblies was further investigated at various $\varphi_{PEO\text{-}b\text{-}PS}$. Taking the morphological evolution of PEO$_{45}$-b-PS$_{16}$ as an example, at $\varphi_{PEO\text{-}b\text{-}PS} = 0$, the binary assemblies of STA and PEO were energetically favored monocrystalline nanoribbons which are similar to the inclusion crystals formed in bulk[23] (Fig. 4g). As $\varphi_{PEO\text{-}b\text{-}PS}$ increased to 25% ($f_{PS} = 1.92\%$), a mixture of nanoribbons and triangular framework precursors was obtained, which also proves the triangular nanoplates are essentially twins composed of different orientation domains (Fig. 4h). At $\varphi_{PEO\text{-}b\text{-}PS} = 75\%$ ($f_{PS} = 5.54\%$), the morphology remains 2D crystalline, but the surface is extremely rough due to the severely inhibited growth in the local height direction. Although the triangular outline is ambiguous, the corresponding SAED pattern and the texture with specific 60° angles displayed on the surface indicate that the assemblies are also twins (Fig. 4i). With the further increase of $\varphi_{PEO\text{-}b\text{-}PS}$ to 100% ($f_{PS} = 7.26\%$), the dense distribution of PS on the surface of the assemblies restrict the 2D growth of crystalline cores, resulting in the formation of 1D nanowires (Fig. 4j). Further results of the CCDSA using other PEO-b-PS with different $\varphi_{PEO\text{-}b\text{-}PS}$ show similar morphological evolution (Supplementary Fig. 14).

All morphological evolution was summarized in the phase diagram (Fig. 4k). The morphologies of the ternary CCDSA will change from a polycrystalline phase (blue region) to a monocrystalline phase (green region) and finally to an amorphous phase (orange region) with the increase of $f_{PS}$. The morphologies evolve from monocrystalline nanoribbons to polycrystalline twins and finally to 1D nanowires with the increase of $\varphi_{PEO\text{-}b\text{-}PS}$ when $f_{PS}$ is low. The size of the crystalline core decreases as the crystallization inhibition was enhanced by increasing $f_{PS}$ and $\varphi_{PEO\text{-}b\text{-}PS}$. In comparison, such a various and tunable morphological evolution could hardly be realized by single-component CDSA of block copolymers.

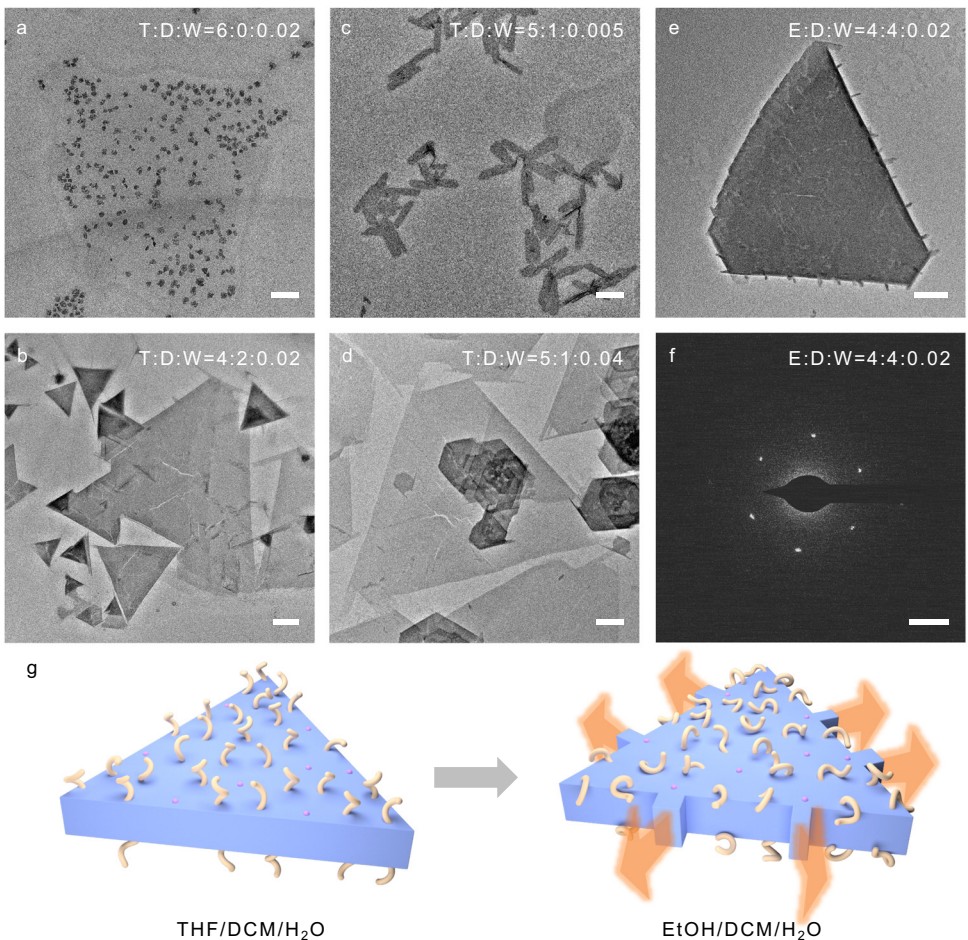

**Fig. 5 | Morphological evolution at various solvent mixtures. a−e** TEM images of STA/PEO$_{45}$/PEO$_{45}$-$b$-PS$_{16}$ in the mixed solvents of different volume mixing ratios. **f** SAED pattern corresponding to (**e**). T stands for tetrahydrofuran, D for dichloromethane, W for water, and E for ethanol. **g** Schematic illustration of flagella-like strips grown from edges of triangular nanoplates due to collapsed PS on the surface in the mixture containing ethanol. The scale bars are 200 nm for (**a−e**), and 0.5 nm$^{-1}$ for (**f**).

## Solvent effects on the morphology

THF is a good solvent for all components, while DCM is a poor solvent for the STA/PEO eutectic, thereby promoting the in situ crystallization in solution. Only small micelles can be formed in the absence of DCM (Fig. 5a). The appropriate volume ratio 5:1 of THF/DCM favors the formation of regular triangular nanoplates. Whereas high contents of DCM are not conducive to the regular growth of the triangular nanoplates because of fast crystallization (Fig. 5b). Less water content only results in small nanoribbons because few STAs dissolved in the solution and cannot effectively participate in co-crystallization (Fig. 5c). The trace amount of water (<1%) is helpful to improve the solubility of STA in the mixed organic solvent. Too much water will lead to thin nanoplates because STAs were solvated and difficult to crystallize (Fig. 5d).

The effect of the protic or aprotic solvent on the morphology by modulating the electrostatic interaction between STA and PEO was also investigated. Ethanol as a protic solvent was used instead of the aprotic solvent THF. The morphology and unit cell structure did not change in EtOH/ DCM/H$_2$O, implying that the organic solvent did not affect the interaction between STA and PEO (Fig. 5e, f). Flagella-like strips could be observed at the edges of the triangular nanoplates, probably due to the different solubility of PS from that in THF-containing solvents. Hansen solubility parameter was used to predict the solubility of polymers in solvents (Supplementary Methods)[51]. Results show that PS is completely dissolved in all the mixed solvents containing THF, while in the mixed solvents containing ethanol, PS is in a critical state as the relative energy difference (RED) is close to 1 (Supplementary Table 3). Therefore, PS enriched on the surface after self-assembly would collapse in ethanol-containing solvents and partially shield the lateral growth front, while the exposed domains without PS continued to grow epitaxially into flagellar-like strips (Fig. 5g).

## Discussion

We propose the concept of ASR and display a novel strategy to mildly induce ASR through the crystallization inhibition effect of short solvophilic segments in CCDSA. The triangular nanoplates triply twinned by orthogonal crystals of $C222_1$ space group have additional triple symmetry. The oriented attachment of homogeneous domains resulting from the crystallization inhibition of the amorphous PS plays an important role in the formation of triangular nanoplates. By modulating crystallization inhibition upon varying the volume fractions of PS, the morphology of CCDSA can be tuned into crystalline 1D nanowires, 2D triangular nanoplates, nanoribbons, platelet, and amorphous reversed worm-like micelles. This work would deepen our fundamental understanding in the mechanism of the novel ASR and symmetry evolution in nature.

## Data availability

All the data that support the findings of this study are available in this paper and its Supplementary Information. Source data are provided with this paper. All data are available from the corresponding author upon request.

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

## Acknowledgements

The authors gratefully acknowledge the financial support from the National Natural Science Foundation of China (Nos. 52073001, 22375009) (J. Z.). We acknowledge the Electron Microscopy Laboratory of Peking University, China for the use of FEI Tecnai F20 transmission electron microscopy. We also acknowledge the Analytical Instrumentation Center of Peking University for the use of AFM.

## Author contributions

S.Y.X. and J.Z. contributed to the conception, design of experiments, drafting, and critical revision of the manuscript; S.Y.X. contributed to the synthesis, analysis, and data collection; W.J.S. contributed to the guidance of structure analysis; J.L.S. and X.H.W. contributed to the discussion of experiment results and revision of the manuscript.

## Competing interests

The authors declare no competing interests.
