## [Peer Review File · Nature Communications]

Apparent symmetry rising induced by crystallization inhibition in ternary co-crystallization-driven self-assemblyReviewers' Comments:

Reviewer #1:

Remarks to the Author:

In the present work, the authors reported the apparent symmetry rising in the co-crystallization of PEO-containing block copolymers and silicotungstic acid. The hybrid co-crystals with orthogonal unit cell exhibited triple symmetry, due to the formation of twin grain boundaries. The authors gave a detailed crystallographic information to the co-crystals. Also, the evolution of morphologies of co-crystals with the component of homopolymers and block copolymers were investigated. The most important part of this work is the mechanism of ASR. The authors stated that ASR of co-crystal is attributed to the inhibition of amorphous PS segments on the crystals. Although crystallization-driven self-assembly (CDSA) has been widely studied in the past decades, the self-assembled structures with ASR is rarely reported. However, before the manuscript can be further considered to publish, some issues in the manuscript should be re-examined and more discussion and interpretation are necessary, in particular, the mechanism of ASR.

1. In Figure 3l, it is shown that the formation of crystalline nanoplates with triplet symmetry (the 'kinetic-controlled metastable state') was due to the stacking or aggregation of several primary nanoribbons. Then, such stacked platelets finally developed to an integral triangular lamella. So, what is the driving force and orientation anchoring of such vertical stacking of primary nanoribbons? As shown in the same scheme, the grafted segment of PS random located on the surface of crystalline nanoribbons, what is the role of PS segments on nanoribbon stacking.

2. Figure 3, after thermal annealing, lots of straight nanoribbons can be observed, beside the triangular nanoplates. It is stated that 'the original regular triangular nanoplates collapsed into nanoribbons, some of which still maintain the triangular framework'. However, a possible case is that these observed nanoribbons were newly crystallized during annealing, rather than disassembling of triangular nanoplates. Note that, in Figure 3i-k, it seems there are structural defects existing the original triangular nanoplates. Therefore, thermal annealing should further improve the structural perfection of crystals, rather than disassembling.

3. It is stated that the triangular nanoplates were 'metastable state', and the straight nanoribbons were in 'thermodynamic equilibrium state'. What means 'metastable state'? Is the crystal form metastable, or it is about the concentration of defects? If the triangular nanoplate possesses a higher (macroscopic) symmetry, why it has a lower structural stability? Again, more evidence is needed to judge the 'stability' of different crystallites, maybe through a DSC measurement.

4. It is better to label the unit cell (ab-plane) in Figure 3k. It will be confused to see 6-fold axes in orthorhombic cell.

Reviewer #2:

Remarks to the Author:

In the field of crystallography, apparent symmetry rising (ASR) is usually limited to metal or alloy materials prepared under severe processing condition. In this manuscript, Zhang et al. reported the crystallization inhibition-induced apparent symmetry rising (ASR) during the co-crystallization-driven self-assembly of PEO, PEO-b-PS and inorganic cluster silicotungstic acid in selective solvents. The authors systematically studied the kinetic-controlled evolution of assembly morphologies and the main factors (e.g., molar ratio of PEO/PEO-b-PS, volume fraction of PEO-b-PS and concentration of cluster silicotungstic acid) influencing the process of co-crystallization. It was found that the solvophilic PS block is key to the ASR phenomenon.

Overall, the experimental results are intriguing and solid. I would recommend the publication of this work in Nature Communications, only after the following concerns are addressed.

1. The experimental procedures are not clearly described. For instance, varying temperature is mentioned in the section of mechanism studies, but it is not clear if this is also used in other experimental procedures?

2. The authors claimed that the formation of triangular nanoplates is a kinetic-controlled process, whereas nanoribbons are defined as an equilibrium product. The result suggests that temperature might play an important role in determining the assembly morphology. What is effect of annealing temperature on the final morphology and ASR phenomenon?

3. The molar ratio of silicotungstic acid to PEO seems to be important to the size and shape regularity of the triangular platelets. What is the exact role of silicotungstic acid in the co-crystallization process and eventually the formation of eutectics? The underlying mechanism should be clarified.

Reviewer #3:

Remarks to the Author:

In this work, Zhang and colleagues report the formation of triangular nanoplates through co-crystallization-driven self-assembly (CCDSA) of an amphiphilic copolymer (PEO-b-PS) and an inorganic cluster (STA). The authors have thoroughly investigated the self-assembly phenomenon and proposed the concept of apparent symmetry rising to achieve these morphologies. I recommend the publication of this study with minor revisions. My specific comments are below:

While triangular shapes are not frequently observed in CDSA experiments, there are some previous reports in the literature. For instance, Kwon and Kim reported the CDSA of PEO-b-PLA resulting in triangular shapes (*Macromolecules* 2021, 54, 10487–10498). The authors should comment/compare their study with the previously reported systems with regard to the assembly mechanism. In addition, the authors should provide experimental evidence to show if the observed self-assembly phenomenon is unique to the selected copolymer or if it can be extended to other systems.

In lines 95-97, the authors mention that assembly was not observed for each individual component in the same solvents, but that does not necessarily indicate that the triangular nanoplates were formed from ternary complexes. The authors should mix the two components (STA and PEO-b-PS) in the same solvents and include the results from those experiments.

In lines 107-112 and Fig. 1d & e, the font sizes in the height profiles are too small to discern clearly. Please reformat the figures to improve readability. From Fig. 1d, the height of the triangular nanoplates exhibits a wide variation. It is important to mention that point when describing the average height. Lastly, considering the substantial variation in height, it is premature to assert that the nanoplates possess the sandwich structure, as depicted in Fig. 1a. Please provide additional evidence to support this claim and strengthen the proposed model.

Regarding Figure 2, the authors should include the simulated XRD profile (Fig. 2e) and SAED pattern along the [001] direction for the STA/PEO crystal structure (Fig. 2f) to enhance the understanding of the crystal structure and support the proposed model.

The authors should test the possibility of varying the PEO length to tune the resulting morphology. This would help discern if PEO90-b-PS could potentially replace the combination of PEO45 and PEO45-b-PS (1:1) and result in similar morphologies.

Reviewer #1

Q1: In Figure 3l, it is shown that the formation of crystalline nanoplates with triplet symmetry (the ‘kinetic-controlled metastable state’) was due to the stacking or aggregation of several primary nanoribbons. Then, such stacked platelets finally developed to an integral triangular lamella. So, what is the driving force and orientation anchoring of such vertical stacking of primary nanoribbons? As shown in the same scheme, the grafted segment of PS random located on the surface of crystalline nanoribbons, what is the role of PS segments on nanoribbon stacking.

Answer 1: According to our understanding, the orientation attachment of such vertical stacking of primary nanoribbons are likely driven by penetration of PEO chains through different domains along the height direction of the nanoplates. The PEO chain can have a variable conformation in imperfect co-crystals of PEO and STA at a low temperature 0°C of sample preparation. The different segments of a PEO polymer chain can participate in different orientation domains under the kinetic-controlled assembly condition without any heating and annealing treatment.

On the other side, as an amorphous segment, PS cannot participate in co-crystallization, and due to its covalent connection with PEO, the co-crystallization of PEO and STA is bound to be limited to a certain extent. Due to the short length of the PS segment ($DP \leq 16$), the co-crystallization between STA and PEO is not completely inhibited, as it is when the PS segment becomes longer ($DP = 82$). The grain boundary defects in pseudo-merohedral twins can provide some space to accommodate amorphous PS without affecting the co-crystallization. In other words, the limited crystallization inhibition effect of short PS segments could also be helpful to induce the formation of twin forms of STA/PEO.

Q2: Figure 3, after thermal annealing, lots of straight nanoribbons can be observed, beside the triangular nanoplates. It is stated that ‘the original regular triangular nanoplates collapsed into nanoribbons, some of which still maintain the triangular framework’. However, a possible case is that these observed nanoribbons were newly crystallized during annealing, rather than disassembling of triangular nanoplates. Note that, in Figure 3i-k, it seems there are structural defects existing the original triangular nanoplates. Therefore, thermal annealing should further improve the structural perfection of crystals, rather than disassembling.

Answer 2: We agree that the nanoribbons could be recrystallized either newly or on

basis of small residual single crystal nucleus from disassembled triangular nanoplates during the annealing treatment. However, triangular crystals are twin crystals with grain boundary defects, which would disassemble at high annealing temperature either 60 or 80°C. Therefore, the thermal annealing can not improve the structural perfection.

To make it clearer, the statement in the main text has been revised as follows: *Afterwards, the as-prepared triangular nanoplates were annealed at 80°C in the sealed solution. After standing for 1 day, the original regular triangular nanoplates almost disappeared, and a large number of nanoribbons can be observed. (Fig. 3f–g). After 3 days, the nanoribbons elongated significantly, and a few of nanoribbons continued to grow on the partially dissolved triangular framework. (Fig. 3h) The higher the annealing temperature, the faster the disassembly of triangular nanoplates. (Supplementary Fig. 8) Annealing at high temperature makes the triangular nanoplates disassembled and slowly recrystallized to form thermodynamically stable nanoribbons with fewer defects.*

Q3: It is stated that the triangular nanoplates were ‘metastable state’, and the straight nanoribbons were in ‘thermodynamic equilibrium state’. What means ‘metastable state’? Is the crystal form metastable, or it is about the concentration of defects? If the triangular nanoplate possesses a higher (macroscopic) symmetry, why it has a lower structural stability? Again, more evidence is needed to judge the ‘stability’ of different crystallites, maybe through a DSC measurement.

Answer 3: Thanks sincerely for your insightful comment. The crystal forms of triangular nanoplates and nanoribbons are the same, that is, they have the same structure of crystal cell. The difference is in the crystal phase: the triangular nanoplates is a twin-crystal phase, while the nanoribbon is a single-crystal phase. In general, when the crystal form is fixed, the single crystal is the most stable crystal phase; whereas the twin crystal has additional grain boundary defects, and its free energy is also higher than that of single crystals. Therefore, we used “metastable state” to describe the twin crystal phase.

The higher (macroscopic) symmetry of triangular nanoplate is an apparent symmetry without considering the microscopic atomic arrangement. Triple apparent symmetry results from the formation of twin phases at the microscopic level, and the cost of apparent symmetry rising is that the nanoplates have more microscopic crystal defects. Therefore, the triangular nanoplate possesses a higher (macroscopic) symmetry but with lower structural stability.

We also have carried out the DSC measurement to investigate thermal behavior of

PEO/STA eutectics. However, we failed to get reasonable profiles (as shown in the following Figure. R1) probably because the melting temperature of the inorganic-organic eutectics might be higher than the thermal decomposition temperature of low-molecular-weighted PEO.

Fig. R1 DSC plots of the PEO/STA cocrystals.

Q4: It is better to label the unit cell (ab-plane) in Figure 3k. It will be confused to see 6-fold axes in orthorhombic cell.

Answer 4: We have modified Figure 3 accordingly by adding ab-plane labels.

Reviewer #2

Q1: The experimental procedures are not clearly described. For instance, varying temperature is mentioned in the section of mechanism studies, but it is not clear if this is also used in other experimental procedures?

Answer 1: The following experimental procedure have been added into the Supplementary Methods section of the Supplementary Information:

A typical Experimental procedure of kinetic-controlled CCDSA: A series of solutions of silicotungstic acid (3 mmol/L) were prepared with THF containing 1 vol.% H₂O and

cooled at 0°C. The polymer mixed solution of $n(\text{PEO}_{45})/n(\text{PEO}_{45}\text{-}b\text{-PS}_{16}) \sim 1/1$ at a total concentration of 2.50 mmol/L in THF/DCM $\sim 3/1$ (V/V) were prepared and cooled at 0°C. A mixture of 100 μL polymer solution was slowly added with 50 μL STA solution at 0°C, stand for 24 h in the refrigerator, and then the self-assembled solution containing the triangular nanoplates was obtained without further treatment.

Experimental procedure of thermodynamic-controlled CCDSA: After the triangular nanoplates were obtained through the above experimental procedures, the self-assembled solution containing the triangular nanoplates was sealed and heated to 80°C for 10 minutes, gradually cooled to room temperature and stand for 1~3 days.

Q2: The authors claimed that the formation of triangular nanoplates is a kinetic-controlled process, whereas nanoribbons are defined as an equilibrium product. The result suggests that temperature might play an important role in determining the assembly morphology. What is effect of annealing temperature on the final morphology and ASR phenomenon?

Answer 2: Thanks sincerely for your insightful comment. Triangular nanoplates were prepared and annealed at 0°C, and defects could be easily formed in the crystallization process at such a low temperature, which may contribute to ASR by formation of pseudo-merohedral twins. (Please refer to Reply 1 to Review #1 and Reply 1 to Review #3 for further explanation)

The effect of annealing at high temperature is to disassemble the triangular nanoplates and slowly recrystallize to form thermodynamically stable nanoribbons with fewer defects. The higher the annealing temperature, the faster the disassembly of triangular nanoplates into nanoribbons. (Supplementary Fig. 8). The triangular nanoplates coexisting with nanoribbons were frequently observed after annealed at 60°C, while the triangular nanoplates have completely disassembled into nanoribbons at 80°C. Accordingly the description and Supplementary Fig. 8 on the annealing temperature has been added into the context of the main text or SI.

Supplementary Figure 8. Effect of annealing temperature on the final morphology. TEM image of incomplete transformation of the triangular nanoplates and nanoribbons at an annealing temperature of (a) 60°C and (b) 80°C. Scale bars: 200 nm.

Q3: The molar ratio of silicotungstic acid to PEO seems to be important to the size and shape regularity of the triangular platelets. What is the exact role of silicotungstic acid in the co-crystallization process and eventually the formation of eutectics? The underlying mechanism should be clarified.

Answer 3: Silicotungstic acid as one of the important components in co-crystal, provides the dominant crystallization driving force for the hybrid system. As a strong acid, STA can protonate the oxygen atoms in PEO, and the STA/PEO forms eutectics in the form of intimate ion pairs through electrostatic interaction according to the Reference 34 (*Chem. Eur. J.* 2017, **23**, 8434-8442).

The effect of the molar ratio of silicotungstic acid to PEO on the morphology of triangular nanoplates have been briefly elaborated in the main text and Supplementary Fig. 14. In general, triangular nanoplates can be formed in a wide molar ratio range, and the size and shape regularity of the morphology decrease with the decrease of the concentration. The crystal structure of STA/PEO co-crystal is not affected by the molar ratio of silicotungstic acid to PEO. The high concentration of STA may lead to the fast crystallization and result in large triangular nanoplates.

Reviewer #3

Q1: While triangular shapes are not frequently observed in CDSA experiments, there

are some previous reports in the literature. For instance, Kwon and Kim reported the CDSA of PEO-b-PLA resulting in triangular shapes (*Macromolecules* **2021**, *54*, 10487–10498). The authors should comment/compare their study with the previously reported systems with regard to the assembly mechanism. In addition, the authors should provide experimental evidence to show if the observed self-assembly phenomenon is unique to the selected copolymer or if it can be extended to other systems.

Answer 1: Thanks for your insightful suggestions. We have found that the triangular nanostructures structures have been reported in face-centered cubic metals, such as gold and silver nanoparticles (*Science* **2001**, *294*, 1901–1903; *Adv. Funct. Mater.* **2006**, *16*, 766–773; *Nat. Mater.* **2007**, *6*, 900–907; *ACS Nano* **2014**, *8*, 5833–5842; *Nano Lett.* **2014**, *14*, 7201–7206; *Nanoscale* **2014**, *6*, 6496–6500). For polymers, as far as we know, triangular shapes currently exist only in the stereo-complex crystals of polylactic acid (PDLA and PLLA), including the mentioned literature (*Macromolecules* **2021**, *54*, 10487–10498; *J. Macromol. Sci. Phys. B.* **1991**, *30*, 119–140; *Macromolecules* **1996**, *29*, 191–197; *Macromolecules* **1997**, *30*, 6313–6322). Therefore, we agree with you that triangular shapes are not frequently observed in CDSA experiments.

Generally, when the polymer chains are arranged in hexagonal close packing to form lamellae, the six crystal faces of $(10\bar{1}0)$, $(01\bar{1}0)$, $(1\bar{1}00)$, $(\bar{1}010)$, $(0\bar{1}10)$, and $(\bar{1}100)$ are equivalent crystal faces, and lamellae tend to form hexagons. However, according to the reference (*Macromolecules* **2021**, *54*, 10487–10498), due to the different configurations of PLLA and PDLA in the PLLA/PDLA stereo-composite crystals, $(10\bar{1}0)$, $(01\bar{1}0)$, and $(1\bar{1}00)$ is one set of equivalent crystal faces, and $(\bar{1}010)$, $(0\bar{1}10)$, and $(\bar{1}100)$ is another set of equivalent crystal faces. When PLA of a certain configuration is slightly excessive, one set of equivalent crystal faces will grow faster, and then develop into a triangle.

The triangular nanostructures, either formed by face-centered cubic metals or PLA stereo-composite crystals, are essentially single crystal nanostructures, but the triangular nanoplates in the present work are essentially a special kind of twins—pseudo-merohedral twins. For pseudo-merohedral twins, the twin operator belongs to a higher crystal system than the cell structure, *i.e.*, a higher apparent symmetry than the symmetry of the structure. The necessary factor for the formation of pseudo-merohedral twins is that there are some special proportional relations in the cell parameters, such as $b/a = \sqrt{3}$ in this work, which is conducive to the approximate coincidence with the original lattice after the 60° rotation. Another factor that induces twinning is the crystallization inhibition of amorphous PS which is covalently bonded to PEO.

Accordingly we have added the above references and the following statement for comparing the assembly mechanism with the previously reported systems in the context of the manuscript: *The triangular nanostructures are not frequently observed in CDSA, and the crystal phase structure is completely different from the previous reported triangles. The triangular nanostructures, whether formed by face-centered cubic*

metals³⁹⁻⁴⁴ or PLA stereo-composite crystals⁴⁵⁻⁴⁸, are single crystal nanostructures, but the triangular nanoplates in the present work are essentially a special kind of twins—pseudo-merohedral twins.

To make it clearer, the strategy of inducing ASR by crystallization inhibition of solvophilic chain segments does not necessarily need to be achieved in a hybrid system of polymers and polyoxometalates, but it requires that crystals can form pseudo-merohedral twins. The ASR phenomenon is universe and could be extended to other system.

In our hybrid co-crystals, replacement of polyoxometalate may fail to form co-crystal or change co-crystal structures (*Nanoscale* **2021**, *13*, 8049–8057; *Inorg. Chem.* **2017**, *56*, 15187–15193); whereas except for PEO, only poly(allylamine) has been reported to form co-crystal with polyoxometalate, but cannot form pseudo-merohedral twins (*Commun. Chem.* **2019**, *2*, 9; *ACS Appl. Mater. Interfaces* **2021**, *13*, 19138–19147). Therefore, we focus on the system of PEO block copolymers and STA.

In addition, research on co-crystallization-driven self-assembly of STAs and PEO block copolymers containing other functional solvophilic segments (*e.g.* luminescent, chiral segments) is under investigation, and the effect of the solvophilic segments on the morphology will be discussed in the next article. In the present stage, we would like to share some preliminary result. We have synthesized the block copolymer PEO₄₅-*b*-PMMA₃₀ *via* RAFT polymerization and then performed co-crystallization-driven self-assembly. The result (Fig. R2) shows a similar morphology of cocrystals to that of PEO₄₅-*b*-PS₁₆.

Fig. R2 The triangular assemblies of PEO₄₅-*b*-PMMA₃₀/STA co-crystals.

Q2: In lines 95-97, the authors mention that assembly was not observed for each individual component in the same solvents, but that does not necessarily indicate that the triangular nanoplates were formed from ternary complexes. The authors should mix the two components (STA and PEO-*b*-PS) in the same solvents and include the results from those experiments.

Answer 2: We have carried out the controlled experiment of two components (STA and PEO₄₅; STA and PEO₄₅-*b*-PS_n), and the experimental results were shown in Fig. 4g, 4j and 4k; and Supplementary Figure 14 and the discussion part of “Morphological evolution tuned by crystallization inhibition”. In brief, as PEO₄₅-*b*-PS_n has a short PS chain ($n \leq 9$), the STA/PEO₄₅-*b*-PS_n can also form triangular or hexagonal crystals. whereas, the STA/PEO₄₅-*b*-PS_n form worm-like micelles as $n > 9$.

Q3: In lines 107-112 and Fig. 1d & e, the font sizes in the height profiles are too small to discern clearly. Please reformat the figures to improve readability. From Fig. 1d, the height of the triangular nanoplates exhibits a wide variation. It is important to mention that point when describing the average height. Lastly, considering the substantial variation in height, it is premature to assert that the nanoplates possess the sandwich structure, as depicted in Fig. 1a. Please provide additional evidence to support this claim and strengthen the proposed model.

Answer 3: The font size in the height profiles in Fig. 1 have been changed accordingly. With regards to the polydispersity of height in Fig. 1d, the results on statistical height of 40 triangular nanoplates were further shown in Supplementary Figure 4. Based on this, the average height is about 50 nm, based on which the sandwich structure was simply proposed to represent the nanostructure.

The polydispersity of height may be attributed to the use of PEO₄₅ homopolymer. When PEO₄₅ co-crystallizes with STA, the crystal surface can continue to grow along *c* axis in the height direction until the surface is mainly occupied by PEO₄₅-*b*-PS₁₆. PS eventually leads to growth inhibition. It is worth noting that the nanoplates does not have the multilayered characteristics with equal interval stepped height, and the surface of some nanoplates is uneven (Supplementary Figure 4). It indicates that the PEO₄₅ segments in the crystal nucleus can be arranged in any form that meets the co-crystal structure, such as folding and interpenetration, and ultimately lead to polydispersity of the height.

Q4: Regarding Figure 2, the authors should include the simulated XRD profile (Fig. 2e) and SAED pattern along the [001] direction for the STA/PEO crystal structure (Fig. 2f) to enhance the understanding of the crystal structure and support the proposed model.

Answer 4: Thanks for valuable suggestions. The simulated XRD profile and SAED pattern along the [001] direction for the STA/PEO crystal structure have been added

into Supplementary Figure 6. The simulated XRD and SEAD results in good agreement with the experimental ones.

Supplementary Figure 6. The simulated XRD profile (a) and SAED pattern (b) along the [001] direction for the STA/PEO crystal structure.

Q5: The authors should test the possibility of varying the PEO length to tune the resulting morphology. This would help discern if PEO₉₀-*b*-PS could potentially replace the combination of PEO₄₅ and PEO₄₅-*b*-PS (1:1) and result in similar morphologies.

Answer 5: We have synthesized a block copolymer PEO₁₁₃-*b*-PS₃₀ by using the commercial reagent CH₃(OCH₂CH₂)₁₁₃OH (PEO₁₁₃) and prepared its hybrid co-assemblies by following similar experimental procedures. TEM images and SAED pattern are similar to those of PEO₄₅-*b*-PS₁₆. These results also have been added into the context of the manuscript:

*Moreover, by varying the length of PEO segment, the hybrid assemblies based on the block copolymer PEO₁₁₃-*b*-PS₃₀ ($f_{PS} = 5.52\%$) show similar morphology and SAED patterns to those of PEO₄₅-*b*-PS₁₆ (Supplementary Fig. 13).*

Supplementary Figure 13. Co-crystallization-driven self-assembly by PEO₁₁₃-*b*-PS₃₀ and STA. (a) TEM image and (b) SAED pattern of the triangular nanoplates assembled from PEO₁₁₃-*b*-PS₃₀ and STA. Scale bar of (a): 100 nm; Scale bar of (b): 1 nm⁻¹.

Additionally, typos and grammar errors have been carefully checked and revised.

Thank you for your time and consideration.
We are looking forward to your positive response.

With all best wishes
Yours sincerely,
Jie

Reviewers' Comments:

Reviewer #1:

Remarks to the Author:

In the present work, the authors reported the apparent symmetry rising in the co-crystallization of PEO-containing block copolymers and silicotungstic acid. The hybrid co-crystals with orthogonal unit cell exhibited triple symmetry, due to the formation of twin grain boundaries. The authors gave a detailed crystallographic information to the co-crystals by using diffraction approaches. Also, the evolution of morphologies of co-crystals with the component of homopolymers and block copolymers were investigated. The most important part of this work is the mechanism of ASR. The authors stated that ASR of co-crystal is attributed to the inhibition of amorphous PS segments on the crystals. The authors provide sufficient details on experiments for the work to be reproduced. Although crystallization-driven self-assembly has been widely studied in the past decades, the self-assembled structures with ASR is rarely reported. This work can give some new insight into the construction and controlling of hybrid nanostructures.

Reviewer #2:

Remarks to the Author:

The authors have addressed all the comments and revised the manuscript accordingly. I am happy to recommend its publication as is.

Reviewer #3:

Remarks to the Author:

The authors have my questions, therefore I recommend publication in its current form.